# Ambivalence towards the Protection of Refugee Children: A Developmental Relational Approach

**DOI:** 10.3390/ijerph19031602

**Published:** 2022-01-30

**Authors:** Jeanette A. Lawrence, Agnes E. Dodds, Ida Kaplan, Maria M. Tucci

**Affiliations:** 1Melbourne School of Psychological Sciences, The University of Melbourne, Melbourne, VIC 3010, Australia; ida@ikaplanpsychology.com.au; 2Victorian Foundation for the Survivors of Torture, Brunswick, VIC 3056, Australia; agnesed@unimelb.edu.au (A.E.D.); tuccim@foundationhouse.org.au (M.M.T.); 3Melbourne Medical School, The University of Melbourne, Melbourne, VIC 3010, Australia

**Keywords:** refugee children, ambivalence, developmental relational theory, co-action, protection, public services

## Abstract

In this paper we analyze the contemporary ambivalence to child migration identified by Jacqueline Bhabha and propose a developmental relational approach that repositions child refugees as active participants and rights-bearers in society. Ambivalence involves tensions between protection of refugee children and protection of national borders, public services and entrenched images. Unresolved ambivalence supports failures to honor the rights of refugee children according to international law and the UN Convention on the Rights of the Child. There is failure to protect and include them in national public services and in international coordination of public health and wellbeing. We identify misrepresentations of childhood and refugeeness that lie behind ambivalence and the equitable organization and delivery of public services for health and wellbeing. With illustrative studies, we propose a developmental relational framework for understanding refugee children’s contributions in the sociocultural environment. Contrary to the image of passive victims, refugee children interact with other people and institutions in the co-construction of situated encounters. A developmental relational understanding of children’s ‘co-actions’ in the social environment provides a foundation for addressing misrepresentations of childhood and refugeeness that deny refugee children protection and inclusion as rights-bearers. We point to directions in research and practice to recognize their rights to thrive and contribute to society.

## 1. Introduction

The future of humanity is tied to the survival and thriving of its children, but that survival is threatened by global and national inabilities to protect, provide for and include millions of refugee children in public services and health care [1,2,3]. In 2013 speaking for the UN, Antonio Guterres described the predicament of millions of refugee children as the potential loss of a generation [4]. That predicament has not been resolved. In late 2021, the UN High Commission for Refugees (UNHCR) reported that there were 82 million forcibly displaced persons of whom 26.4 million were designated refugees with half of them (52%) children under 18 years of age [5]. The COVID-19 pandemic has exacerbated the trauma related to homeland disruptions, insecure transitions and resettlement stresses of refugee children and their families [6]. By international law and global agreement, refugee children have legal rights to protection, provision and inclusion. These rights are guaranteed by the 1951 Refugee Convention [7] and are articulated in detail in the UN Convention on the Rights of the Child (CRC) [8,9]. The UN specifically included global health, wellbeing and equality in the Sustainable Development Goals for 2030 [10]. However, coverage and equity in public service and health care are undermined by the upsurge in forced migration, organizational challenges for the World Health Organization and nation-based tension between international obligations to protect refugee children and the politics of protecting national sovereignty and population wellbeing [3,11]. As Gostin [3] observed, although human rights law guarantees nondiscrimination and health care, migrant health needs and provisions are widely disregarded. That tension is part of the contemporary attitude of ‘ambivalence’ that according to Bhabha has become a generalized twenty-first century response to migrant children ‘on the move’ [12] (p.5). Children’s migrations change and shape the world and bring out unresolved issues about children’s protection and vulnerability at all levels of social discourse. They also place rights-based obligations on transition and resettlement countries to honor the CRC, Article 22 mandate that refugee children be treated the same as citizen children [8]. The protection and inclusion of refugee children should be major policy considerations for the organization and delivery of public health and services. Public health policy fails to reconcile internationally legislated rights with political and economic concerns whenever a society vacillates between attending to the humanitarian obligations to protect vulnerable children who are ‘human like me’ and instead protects the state and its members from outsiders seen as threatening ‘others’ who deserve punitive rather than protective treatment, because they are ‘not like our children’ [12] (p. 11). For example, many nations concentrate on screening refugees for communicable diseases instead of including them in comprehensive, population-wide health systems [2]. Thus, the prevailing ambivalence called out by Bhabha is an impediment to inclusion and equality for a significant section of humanity.

In this paper, we analyze the oppositional ambivalent attitudes by which protection is both directed towards and away from refugee children [12,13,14]. The tension generated by simultaneously holding instead of resolving such oppositional attitudes is fueled by misrepresentations of the experiences of childhood and refugeeness [15] that fail to recognize refugee children as active participants in the sociocultural environment. The consequences of such misrepresentations include hesitancy to meet protection obligations and ineffectiveness in integrating, organizing and delivering equitable public services and health care across and within jurisdictions [1,3]. 

We propose a developmental relational understanding of child refugees that positions them appropriately as participants in the sociocultural environment and provides a basis for resolving misrepresentations that fuel ambivalence. We first describe a developmental relational framework that positions refugee children as active participants in social encounters. We then identify the dimensions of twenty-first century ambivalence towards the protection of refugee children and specify how ambivalence impedes their inclusion in public health systems. Contemporary developmental science provides patterns of evidence to undergird a rights-based approach to protection by treating refugee children as co-constructors of situational interactions and developmental trajectories [16,17,18]. Honoring refugee children’s right to protection depends on understanding their patterns of interaction with people and institutions in a variety of situations and over time. We show with empirical examples, how refugee children are dynamically interactive in specific situations. A relational and developmental account of their interactions provides a foundation for a rights-based approach that protects their right to thrive as they develop.

## 2. A Developmental Relational Framework

A relational framework as generated in developmental relational system theory [16,18] positions each refugee child as an embodied, living system (person) connected to other organic and structural systems within the total ecological system. Living systems are interrelated, so that interactions between children, adults and social institutions affect how each of them functions and develops. People’s situated experiences are neither wholly shaped by the environment nor by their own activities but are ‘co-constructions’ brought about by multiple system forces that work ‘co-actively’ [17,18,19]. Persons engage in reciprocal, bidirectional actions that by their co-actions contribute to adaptation in themselves and others. By their interactions, people also change the sociocultural environment itself, such as when they routinize and ritualize existing practices (e.g., food taboos) and introduce new practices (e.g., adopting food or communication norms of an entering culture). 

Within the ecology of a specific situation, environmental and social forces enable and constrain the expression and development of any child’s thoughts, feelings and behaviors by the way life is organized and how people relate to that child’s activities [19,20]. Each child in turn constrains and enables other people and institutions by their actions to resist or modify adult activities. Children are active, but they are not the only actors, and along with all actors, they construct personal meanings for the interactivity [21,22,23].

Children may be protected from harm or be placed at greater risk by the way that they and others interact, and children, like all persons, may act to change the dynamics. For example, when adults fail to provide children with safety and protection, children may seek outside help or simply remove themselves from the aversive situation. Street children in Accra, Ghana told researchers that they decided to leave their homes and live on the streets because the people at home failed to care for them. They claimed they were taking independent action, while acknowledging that they acted because of environmental pressure [24]. Failing to recognize the reciprocal interactivity of refugee children in contextualized activities leads to misunderstanding and ambivalence about their position in sociocultural life.

## 3. Ambivalences towards Protection of Refugee Children

Protection is an activity providing safety and a shield from risk and threat for oneself or another person. It is a fundamental human activity generally seen as the responsibility of more powerful people towards the less powerful at all levels of human experience. Responsibility to protect children has been a social ideal at least since the industrial revolution and the enlightenment. Hämäläinen [25] traced the history of how concern about protection led to the formalization of children’s rights. The human propensity to both mal-treat and shelter that Hämäläinen observed reinforces the need for a rights-based approach. That dual propensity may be most prominent in the forced return of refugee children to unsafe countries as practiced across Europe with seemingly little concern about what will happen to them [26]. The alternative to a rights approach, a welfare approach, relies on benevolence that denies children their position as active, developing persons and depends on individualized or group acts of benevolence, and does not have a public basis for the administration and assessment of children’s protection. A rights-based approach lays out the general principles for ethical treatment across a nation’s population [27]. These general principles are translated into public policies and systems by legislation that then are applied to the circumstances operating in specific situations. Even when rights are not honored, there are legal avenues for confronting inequity and for making claims and appeals. Ambivalence about the status and claims of refugee children is important because it can disrupt the processes of instantiating their rights to protection at different levels of public provision. 

### 3.1. Protection of Refugee Children

Protection of sovereign states and their interests may sideline the obligations of a government and its population to honor the rights of refugee children. Rigby et al. [28] subjected one UK policy to close documentary analysis. They uncovered the discriminatory concepts and inferences used in a policy purporting to safeguard unaccompanied child asylum-seekers. Concepts describing the problem of unaccompanied asylum-seeking children set up false dichotomies between children who were judged ‘deserving’ or ‘undeserving’ of protection. Words such as ‘clandestine’, ‘dangerous’, and ‘choice (making)’ justified reducing protection for this needy class of refugee child. Inferred meanings cast unaccompanied children as objects of risk to themselves and UK citizens, and as threats to public safety who were seen as either incapable or unwilling to manage their own risk. Ulrich and Crosby [29] recently confronted the inequity of the Title 42 immigration policy in the USA. Title 42 purported to address public health issues by limiting the spread of COVID-19 by expelling, without resource to appeal, refugees from five countries (Mexico, Guatemala, Honduras, El Salvador, Haiti), although there was no scientific evidence that they presented a public health risk. A senior official resigned in protest of this violation of international obligations to refugees. 

Discriminatory policies adopted across Europe and North America are exemplified by immigration procedures Derluyn identified in Belgium [30]. Along with other European governments, Belgium created special care and reception facilities for unaccompanied minors deemed as more and less vulnerable. Structures and rules were geared to supposedly objective signs of the children’s vulnerability, and these signs were used to provide extra resources and support for a subset specified as deserving. Refugee children who were female, under 14 years of age, or previously diagnosed as exhibiting mental health issues were judged to be ‘extra vulnerable’, and so deserving of the kind of care given Belgian children. Children not meeting those criteria were categorized as less or not vulnerable, and by implication as undeserving. 

Classification by vulnerability is not confined to the Belgium system but has been used informally by migration officers and social workers as well as formally across systems. There is some evidence that stereotypic images of vulnerable, legitimately needy children give workers quick ‘how to recognize’ criteria to use along with informal yardsticks of age and social competence. Migration workers felt they could detect ‘fake children’ who were older than 18 years or more competent than local children [31,32]. Derluyn [30] interpreted this kind of processing regime as a migrant managing scheme that reduced cost, but by objectification directly opposed the CRC ‘best interest of the child’ principle. The Rigby et al. [28] and Derluyn [30] analyses highlight the ambivalence displayed by national bodies and institutions where claims for protection from refugee children (voiced or silent) are pitted against other claims. 

### 3.2. Protection of Refugee Children versus Conflicting Claims

Conflicting claims for protection include protection of national borders and sovereignty, institutional resources and culture, and the ideological preservation of images of governments and populations as benevolent protectors of the worthy.

#### 3.2.1. Protection of Borders and National Sovereignty

Some countries have applied the deservedness motif and actively opposed the protection of refugee children against the protection of national borders, sovereignty and the domestic population. For instance, Australia has persisted with the contrast between deserving and undeserving refugees, specifically in terms of how they arrive in the country. Successive governments (Liberal and Labor) have baulked at providing protection depending on whether refugees dared to arrive uninvited by boat. Boats meant threat. The Liberal (conservative) prime minister John Howard in his 2001 election speech in relation to national sovereignty and security twice declared: 

We will decide who comes to this country and the circumstances in which they come…. We will defend our borders and we’ll decide who comes to this country. But we’ll do it within the framework of the decency for which Australians have always been renowned [33].

Writing in 2013, Crock demonstrated how the Australian government embellished the boat threat [34]. To justify its failure to honor the protective rights of actual children, it claimed to be saving putative children who risked drowning if they came by boat. Shifting the protection discourse onto putative children enabled the government to maintain a presumed humanitarian rhetoric to justify a punitive, deterrence goal that it presented as protecting national borders. Such opposing policy positions and practices seriously question Australia’s honoring of internationally agreed protection obligations [35]. 

A similar approach to so-called ‘illegal’ refugees, according to Mousin [36], shifted the US policy debate away from US law and CRC guidelines, with the consequence of denying refugee children the right to legal representation and subjecting them to lengthy periods of family detention or family separation. Detention of asylum-seeking families and their children has become common practice in Europe, the USA, and Australia [31,35,36,37]. 

Populations can be as ambivalent as politicians in their attitudes, swayed by graphic media images of vulnerable children, border protection rhetoric and forecasts of economic disaster. One in four Australians supported the turn back of boats in 2013–2015, while at the same time there was majority population support for the country’s humanitarian refugee program [37]. Throughout Europe, minor parties gained electoral success on issues of immigration and race. Anti-immigrant sentiments rose in relation to European perceptions of deteriorating economic conditions, but as Kuntz et al. [38] revealed, the economic gloom came not from actual data, but from popular perceptions, with economically deprived populations typically holding more xenophobic attitudes towards refugees.

Ambiguity and tension filter down from macro-political and population levels into the administration and delivery of health and education to children seeking asylum. Ottoson et al. [11] found that children’s case workers in Sweden had difficulty in interpreting inconsistent policy priorities when organizing health provisions for refugee children and managing restricted resources on the ground. Competing concerns, demands and resources produce the kind of policy and practice blind spots that Vathi and Richards [26] also found in the organization of migration and child protection services in Albania that meant no assistance was offered to either refugee children or Albanian children returning after periods of migration in European countries. Social workers had to navigate through ambiguous welfare systems at the same time as following migration management laws. School teachers were left with little awareness of the impact on children of the discords between government and educational systems. Teachers are not immune from stereotypes. In Germany, pre-school teachers who were operating from negative stereotypes of refugee children also said they experienced more difficulties with refugees than with other pupils, including hyperactivity and inattention [39]. 

Recent structural moves to greater control of migration and associated economic measures have produced difficulties for UK secondary schools in enrolling and educationally supporting asylum-seeking children [40]. An official ‘dispersal’ policy meant that children and families were distributed across UK cities with limited resources. Head teachers at local schools had to navigate through competing needs of refugee and local children. They expressed concern about the inequities of how asylum-seeking students were admitted to schools and the burdens involved in keeping track of the children’s progress when inadequate funding left schools to compensate for gaps in national provisions. A similar migrant dispersal policy provoked xenophobic demonstrations by parents in some Greek schools [41]. Parents’ objections were intense in wealthier middle-class areas, although their fears and objections were actively countered by official and community social justice initiatives.

#### 3.2.2. Protection of Entrenched Images

Protection is disrupted when conflicting claims reach deep into people’s ideological imagery and stereotypes. Biases are fed by the juxtaposition of positive local images and negative refugee images [42].

*Positive Domestic Images.* The self-image of a benevolent protector of innocent child victims is opposed to the self-image of one obliged to protect the rights of refugee children. Living with the dissonance incurred by not scrutinizing competing views of the state, its citizenry and its asylum seekers avoid confronting the rights-based position accorded to refugee children in international law and conventions. People and institutions go to great lengths to preserve positive images of their own benevolence and philanthropy. Politicians are adept at promoting their positive images, and the media is adept at nurturing similar self-images in the population [13,42].

Crock’s uncovering of the Australian government’s contrast between its responsibilities to actual and putative refugee children also uncovered the government’s reliance on its self-preserving images as justification of decades of harsh offshore processing and detention of refugees [34]. This self-protective and self-justifying positioning was subjected to global scrutiny along with another incident that was used politically to reinforce alienating and distancing ‘we/other’ imagery of undeserving refugees. In support of its hardline exclusion of boat people, the Howard Government nurtured rumors that refugee parents on a sinking boat had thrown their children into the sea so they would be rescued by the Australia navy. Slattery’s analysis of media reports revealed how this unchecked fabrication gained political mileage for the government allowing it to present ‘irregular’ or ‘illegal’ refugees as people not deserving benevolence or protection [43]. 

In the UK, popular media demonized refugee parents as abandoning their children. Media photographs of children in Calais camps that rarely included parents did include props such as teddy bears signaling childhood innocence. These images permitted the British public to hold on to the image of itself as protector and substitute caretaker, taking up moral duties on behalf of lone, abandoned children [13]. 

*Negative Images of Refugee Children.* Distorted images of governments and citizens gain from being paired with a one-dimensional image of refugee children that misrepresents the complexity of their refugee experiences. The misrepresentation is particularly acute when it focuses solely on vulnerability as in McLaughlin’s analysis [13]. When that image could not be sustained, the media and the population reconfigured the image of the British public as the ‘good one’ taken in by ‘fake child refugees’ [13] (p. 1763). Face-to-face encounters and real photographs of children who arrived from Calais allowed people to replace one simplistic, distorting image with another. Framed by stereotypic views of juvenile delinquents, vigorous adolescent bodies did not match the constructed image of helpless children but supported the alternative interpretation that these were adults (older than 18 years) posing as children, or blamable delinquents without proper documentation. Calls for re-assessing children’s ages revived mechanisms for objective age assessment that had been directed at other cohorts of refugee children stigmatized as ‘imposter children’ or ‘bogus refugees’ by virtue of their appearance, independence and self-management skills [32]. 

Biased print and social media sustain misrepresentations and add to the distress of asylum-seeking and resettling groups. Patil and McLaren analyzed texts of articles and letters in five major Australian newspapers over a 12-month period [42]. Asylum-seeking children were packaged with people smugglers and ‘rule breaking’ parents in discourses of illegality and fear. The children were represented as not deserving freedom and normal life. 

Representations of children as dependent, compliant, and passive victims have been exposed along with another Western assumption that normal children’s lives are always localized and stable [44,45]. Derluyn and Vervliet [46] directly confronted the tendency to question the normality of unaccompanied refugee minors. Instead, they attributed abnormality to situations, not children; “These children are not inherently vulnerable to developing mental health problems, but (that) the situations they are in, now and in the past, are likely to evoke this vulnerability” [46] (p. 106). 

In summary, allegiances to national interests or to entrenched images motivate governments and individuals to hold in abeyance their obligations to honor the rights of refugee child rights-bearers. Refugee children may be framed as threats to national sovereignty, local homogeneity and prosperity or as distortions of normal childhood. Stereotypic images of deserving and undeserving refugee children do not resolve the protection question, but instead support unresolved ambivalence. Even when the one-dimensional image of the innocent, passive child victim is revealed as counter-factual [13], that image seems to be retained as one competing image precisely because it matches romantic Western assumptions about normal childhood [44]. 

Conflicting one-dimensional images of refugee children sustain ambivalent attitudes and disrupt protective processes by their assumptions about vulnerability and normality. Their persistence also exposes misunderstandings of childhood and refugeeness and how these experiences are intertwined in refugee children’s interactions with the sociocultural environment. Children are vulnerable, but they also are agentic, and vulnerability and agency can be intertwined in people’s thinking and actions [47]. One-dimensional perspectives do not reckon with the contextualized interactions through which developing persons of all ages relate to others by their reciprocal, bidirectional activities [17,18,19]. Refugee children interact daily with family, teachers, community members and authorities. Their perceptions and meanings for their encounters are expressed in actions that are reciprocally constraining and enabling. Children contribute to adaptations in the situation, other persons and themselves. With varying forms of support, children must navigate their way through local beliefs and practices that have varying proximity to their heritage cultural beliefs and practices. For example, when families are resettling in a new country, children typically become acculturated more quickly than their parents, acquiring local language and customs through school and friends. They often become cultural bridges and family interpreters, and by their encounters open up family power dynamics to change [22,48,49].

## 4. A Relational Positioning of Refugee Children

A developmental relational systems approach positions refugee children as contributing participants in the ecological system. They are neither passive victims completely under the control of external forces, nor independent non-contributing observers. A refugee child is one embedded living system who relates to other systems by their thinking and acting. Relations between people and social systems involve dynamic interactions of different intentions and influences that by their interactions bring about change in the larger social system as well as in the individual person within it. Open systems, whether organic or structural, are open to each other and to the possibility of promoting specific experiences and also flow-on effects [15,16]. For example, length of time in assessment centers, insecure residence and family separation have flow-on effects on refugee individuals’ mental health whether they be parents or children [49]. Patterns of exclusion of refugees from health services also have flow-on effects to other systems of service for them and for the total population [50].

There is general evidence of children’s increasing influence on family dynamics as they develop [51]. Children give new meanings and uses to adult ideas and skills and find and exploit loopholes in adult rules [21,22]. However, the contribution of refugee children to social change has not been a research emphasis in developmental or other social sciences. Accepting the victim image reduces expectations of refugee children’s agency and restricts how their interactions are observed or interpreted [21,52]. Being cast exclusively as a victim may be no more conducive to searching for a child’s influence than being cast as absent or an appendage [15]. There are significant examples, however, of refugee children’s navigation of sociocultural situations that, within different situations for different children, have involved their use of: silence or obfuscation of details about their lives; resistance to adult arrangements; or their contributions to community projects and concerns. 

### 4.1. Using Silence and Obfuscation 

Failure to reckon with refugee children’s input is partly because other people’s blinkered, entrenched images cloak children’s contributions, but also partly because some children actively divert attention from their circumstances. In situations of danger, children have actively camouflaged their identities and abilities by being silent or selectively hiding their identity and activities [53,54,55,56,57]. For example, children seeking asylum in the UK actively presented care workers and immigration officials with silence or duplicity about their documentation and their family circumstances and decisions. Children had a range of purposes, including coping with the system, resisting control of their lives, distancing themselves from the asylum-seeker label, stigma, and teasing, and, for themselves, focusing on forgetting the past and moving on [53]. 

In another context, children and young people aged between 12 and 22 years described how they hid their identities in Ugandan society following civil war. Born of conflict-related sexual violence, they had lived most of their childhood in the camps of the Lord’s Resistance Army. When the war ended, they were stigmatized and had to grapple with the meaning of being born and raised in captivity and the changes required for life in regular society [55]. They made efforts to build false identities that allowed them to belong while remaining aware of their past lives in the bush. Some felt they were losing as well as gaining freedoms, with girls now feeling they were expected to do household chores that had not been part of their camp life. 

There is little evidence of the effect of children’s silence or obfuscation on the wider community. However, O’Higgins [52] noted that refugee youths had better outcomes with their social workers in the UK if they did not reveal their independent activities when those activities, such as independent travel or negotiations, did not conform to the social workers’ assumptions of dependency and vulnerability.

### 4.2. Resisting Adult Arrangements

Some children have taken advantage of opportunities to assert themselves and their causes. When cultural elders of Congolese refugee groups failed to take leadership roles in Kenyan camps, young people moved into the gap and took up leadership roles for themselves [56]. Other children became accustomed to making independent decisions for themselves and others when they were forced to become heads of families in times of disaster in Zambia [57]. They were reluctant to give up the power they had developed. One overt but unexpected example of taking control of events involved unaccompanied girls who were awaiting asylum decisions in Finland [58]. Having previously exercised adult responsibilities, many were pleased to have a reliable adult to make decisions. They felt the professionals knew what was best for them, but they did want to be informed about decisions affecting their lives. They sought a balance between protection and participation rather than to take power away from trusted adults.

Relations with adults may require more than individualized actions, and some children have built up collective actions that contain strategies for others to pick up and use. For instance, Thomson [54] found that Somali girls developed an underground network to assist them in navigating interactions with different men in Eastleigh district, Nairobi. Girls between 13 and 19 years generated and disseminated multiple tactics for dealing with men, including family members, neighbors, and Kenyan police. Secure relationships with other girls across the network facilitated accessing and practicing how to use silence and muted voice to preserve their sense of self and personal power in interactions with authorities and sexual predators. By contributing to the network and by their modified social behaviors in the wider community, network members successfully maneuvered around powerful and predatory men and left traces of effective strategies in the environment for other girls to use.

Veronese and Cavazzoni [59] specifically investigated the agency expressed by Palestinian children in the Dheisheh refugee camp. They focused on the children’s interactions with family and community that influenced their wellbeing and development. Twenty-nine children ranging in age from 7 to 13 years were asked to draw a map of the camp and to color in green places where they were safe and in red, unsafe places. A subgroup walked researchers through their maps explaining what the places meant to them. Despite recognizing the dangers associated with the streets and outdoor spaces, children used these risky spaces as sites for play and working through the tensions between constraints and freedom. Girls were intent on reclaiming the streets and outdoor places for themselves and other girls who might otherwise be restricted by traditional constraints on females.

In an earlier Palestinian study, Marshall [60] used multiple methods including discourse analysis and children’s drawing, mapping, photography, and participatory videos to research the outcomes of humanitarian programs for children in a Balata refugee camp. He focused analyses on children’s ways of being political and their responses (conformity or resistance) to framing their experiences within a trauma discourse. NGO humanitarian organizations designed programs of healing, art therapy and internet use to directly respond to children’s trauma-related behaviors and to improve their mental health by introducing alternatives to violence. However, the children had their own ideas and saw themselves as active members of a community in resistance rather than as victims. They took up the tools designed to treat trauma and used them for the politics of anti-occupational resistance. For example, one group took photos and videos in support of commemoration of martyrs and care for the local people. One girl took photographs to encourage the children in America she thought did not have the strong family and society they had in Palestine, “when they see what we are doing, they will be inspired to be strong” [60] (p. 292). The children deliberately refigured the intervention strategies provided by humanitarian workers. Marshall concluded that by focusing on individual symptoms of trauma, humanitarian organizations depoliticized the occupation, and by pathologizing the Palestinian children, limited the scope of their political subjectivity.

One of the clearest examples of intent to manipulate social interactions was named ‘victimcy’ by Utas [61,62]. Victimcy describes the intentional manipulation of images by self-presentations aimed at achieving a particular social end. Utas applied it to self-staging by child soldiers and exploited women and girls to present themselves as victims of war. Child soldiers used this self-presentation strategy in their post-war narratives to family, journalists and authorities to reconstrue their war experiences so they could dispel blame and assist their reintegration into society and post-war livelihoods [62]. This micro-level manipulation of images and information, Utas argued, needs to be analyzed in relation to macrostructures in which vulnerable people interact with the wider society as well as with individuals.

### 4.3. Being Part of Community Projects and Action 

The Veronese and Cavazzoni [59] study reported how children actively sought to connect with the community around memorials. Another study by Veronese et al. [63] investigated the ideas about their agency and social involvement expressed by Palestinian children aged between 6 and 15 years who were living in one of three camps. They asked the children to write an individual self-characterization in Arabic and to draw a portrait of themselves, their family, and scenes where they saw themselves as playing an active part. With situational differences according to camp, Palestinian National Identity was observed in 19% of written narratives (17% of drawings) from 122 children. Political agency in the form of civic engagement and activism was a theme in 15% (16% of drawings). Children expressed being and feeling to be a member of the Palestinian community and an active participant in the struggle.

A particular war-related situational problem that could only be resolved with action by children arose during the siege of Sarajevo in 1992. Lucic [64] later interviewed young people who had helped change the fortunes of the besieged city. The army had dug a tunnel under the airport to transport food, goods and military equipment in and out of Sarajevo, but some parts of the tunnel were too small for adults to negotiate. Orphaned adolescent boys were enlisted as regular recruits by the army to carry packages through the tunnel, and then as irregular carriers for smugglers. An underground economy to service the citizens evolved by courtesy of children. Children found themselves called upon to make decisions and to exert their strength for the common good. Life was changed for them and for the adults who were dependent on them for that explicit activity. Lucic [64] reported how the children’s lives were transformed and how the skills and knowledge they developed during the war shaped the progress of their lives across two decades. As their society was changed by the war, their contribution became part of a lasting narrative.

Overall, these illustrative studies indicate that children are capable of positioning themselves squarely in relationships and activities that are important to them. The studies showed how refugee children generated a range of ways to act upon and maneuver through the actions of other people. While some children used silence, obfuscation or false self-presentation as means of self-protection or gaining some control over events, others joined political movements or underground networks of resistance. They interpreted prevailing constraining and enabling forces from their perspectives and responded to other people’s perspectives and intentions with ingenuity—sometimes conforming, sometimes resisting.

To characterize refugee children solely as passive victims or as both victim and threat are two perspectives that sustain ambivalent, self-serving attitudes. However, these positions also overlook children’s contributions to the co-construction of situated events. The form of refugee children’s contributions to such events along with their individualized interpretations may be as varied as the settings in which these children make their mark and are marked by those events. In all these studies, community reactions to children’s acts are mostly missing. Short-term and especially long-term outcomes are not reported. This is understandable, since none of these examples involved specific analysis of the co-active construction of events by children and others that belong to a relational framework for events. 

### 4.4. Relational Interactivity

The developmental, relational approach we described recognizes refugee children as natural, embodied contributors to different forms of relational encounters that occur in the various situations of their refugee experiences. Their participation, although often unacknowledged and unseen, is one organic part of a larger open system in which it and other parts contribute to change in the dynamics of the immediate encounter and contribute to change in the social system [17,19]. Immediate, contextualized and longer-lasting developmental changes may emerge in participating children and adults at neurological, cognitive, emotional, behavioral or even genetic levels of their organic functioning [17]. Organic change involves adjustment, for example, when children read situations as negative and generate new communication or management skills to respond to political or local cultural constraints and affordances. Relating involves the bidirectional, multi-level actions and effects for all participants. The form and direction of change are partly due to individuals, and partly to ecological and systemic factors and their coming together so that change is probabilistic and truly co-actively constructed [17,49].

The environments of children’s interactive participation are never neutral spaces. The physical and sociopolitical terrain constrain the opportunities and resources that refugee children may access and use, for instance, as when public service resources of resettlement countries are limited and cause governments to be less generous to migrants and refugees [1,3,50]. 

Refugee children may accommodate to physical, cultural and interpersonal constraints or they may resist and disrupt those constraints, turning them into opportunities for pursuing their intentions [61,62]. It is not surprising that some refugee children are wary of adults in authority and prevaricate about revealing their experiences and feelings [53,55,62]. It also is not surprising that people in resettling countries, when challenged by the volume of asylum-seekers, fall back on entrenched images that allow them to overlook the contributions that refugee children can and do make. Due recognition of the reciprocal interaction of refugee children with people and institutions paves the way for redressing misrepresentations that fuel ambivalence and provides a foundation for resolving ambivalent attitudes that stand in the way of supporting the rights-based thriving of refugee children. Acknowledging the interactive participation of refugee children in sociocultural events indicates that they are rights-bearers whose rights for equal protection and developmental thriving have priority over competing calls to protect national borders and entrenched interests and images [8,27].

## 5. Conclusions and Directions

We began with the importance of positioning refugee children appropriately in global and local contexts, locating them as *co-active* rights-bearers. Their legitimate claims for protection and inclusion in public service provisions demand recognition and action by social systems at all levels of their interactions with child migrants. Despite the legitimacy of their rights, the ambivalence that Bhabha [12] exposed leaves the protection of refugee children unresolved. Oppositional priorities give rise to inequities in the public organization and delivery of health, education and welfare [3,50]. Children’s protection is especially neglected where it is, or seems to be, opposed to the protection of national borders, local interests, and entrenched attitudes. In the face of international laws and guidelines, the persistence of conflicting calls for protection is fueled by misrepresentations of childhood and refugeeness. Central to dispelling such misrepresentations is the developmental understanding of refugee children’s systemic interactions with the sociocultural environment. The form and content of their co-activity is neither solely as a responder to external forces, nor solely as an interpreting mind—both positions that have been embraced over the history of psychological thought and led to misinterpretations of children’s position in society [19,23]. 

As Bhabha [12] pointed out, to expose ambivalent attitudes does not automatically resolve the ambivalence. Resolution of ambivalence to providing protection for refugee children requires a categorical shift towards translating legal principles and guidelines into policy and practice. The crisis for future humanity that Guterres identified demands action to reposition refugee children as interacting participants at all levels of contemporary life [4]. To achieve such recognition and repositioning rests on developing a clear understanding of how refugee children interact with the sociocultural world—what their co-constructive actions are like and what their reciprocal co-actions with others produce.

The passive vulnerability motif is dispelled by the ability and willingness of these normal children to interpret, navigate, and manage their abnormal refugee situations [46]. Children have the ability and propensity for engaging with the sociocultural world with varied interpretations and meanings and are capable of both initiating and responding to relational encounters. Their protection demands more than locking them out of activities and decisions like non-acting exhibits. It demands engagement.

*Directions* for repositioning refugee children as rights-bearing participants is dependent on at least two forms of developmental evidence: micro-level analyses of the dynamics of situated interactions and longitudinal analyses that track emerging and long-term changes in children, adults and systems.

Micro-level analyses of specific interactions can reveal how refugee children intellectually and emotionally adjust to specific contexts by negotiation or resistance, that in turn elicit negotiation, resistance or accommodation by adults [21]. Understanding of children’s abilities can shape public health messages directly for them, especially where they have greater language and cultural skills than their parents [22], as for example, in the distribution of infection information and prevention and vaccination provisions.

Evidence of children’s co-actions in engaging with people and institutions is thin at this stage. Researchers have not specifically focused on the interactivity of refugee children’s social encounters. Program developers and practitioners have not always understood the power of children’s motivations and actions [27,52,53]. We know little of the processes of their co-actions, and less of the immediate and long-term outcomes of their engagement with authorities in schools, communities and legal jurisdictions. 

Longitudinal evidence of individual children’s experiences is needed to reveal adjustments in emotional, intellectual or neural functions and the long-term effects of these adjustments over the life-course. Frounfelker et al. [49] pointed to the lifelong mental health risks associated with individualized refugee experiences and also the fall-out for them of the experiences of other family members. We agree with Frounfelker et al.’s recommendation that long-term effects of refugee experiences be traced in epidemiological and family-based longitudinal analyses as a basis for intervention programs. 

The examples we highlighted point to children’s contributions to political resistance, particularly at the macro-level of solidarity with community struggles [53,60,64] and at the micro-level of resisting or subverting adult arrangements in migration and educational facilities [21,56]. Corsaro’s [21] reflections on his earlier observations of children’s subversion of adult authority also is a reminder that researchers bring their own meanings to the interactions they observe, and that those meanings are open to development and modification over time. 

There now is a way forward to resolving the ambivalence towards refugee children and their right to protection. Acknowledging their active participation in the sociocultural environment presents possibilities to produce thick evidence of their relational contributions as dynamic interactions with people and institutions. These children are survivors and have already demonstrated their propensity for recovery and resilience across situations [48]. Their perceptions and meanings are valuable elements of how their rights can be implemented [27]. The effects of their co-constructive work need to be closely observed in situ and ontologically over time. The developmental relational approach directs research to track how individuals and institutional systems respond to children’s initiatives and reconfigurations of adult intentions and arrangements. Protection and the opportunity to thrive and develop in society are the rights of refugee children, and a major responsibility of people who interact with them at every level in every situation. Recognizing and facilitating children’s interactivity with people and institutions paves the way for supporting the thriving and future of refugee children and of humanity.

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
