# Peer review of "Ambivalence towards the Protection of Refugee Children: A Developmental Relational Approach"

_ijerph, 2022, doi:10.3390/ijerph19031602_

Round 1
Reviewer 1 Report
Aims: This paper aims to analyze how international ambivalence between protecting refugee children as passive victims vs. protecting national borders has led to failure to honor children's rights to the protection mandated by the United Nations Convention on the Rights of the Child and maps misrepresentations of childhood and the refugee experience behind this ambivalence. To address these issues, the authors propose a developmental relational approach that redefines child refugees both as agentic in interacting with people and institutions and as rights-bearers. The authors cite illustrative studies to guide research and practice consistent with this reframing.
Strengths: This paper is an important and timely overview, critique, and reframing of key gaps in global public discourse, policies, practice, and research on refugee children. It is exceptionally lucid and cogently written, and it draws on an importantly broad and current set of international evidence.
Challenges: Beyond asserting the need for a "developmental relational model", with evidence involving descriptions of refugee children as a group (Lucic, Utas, Veronese), this reviewer did not find concepts or empirical evidence to bring this claim into sufficient developmental analysis. For example, a classic set of developmental questions (Wohlwill, J. F. (2016). The study of behavioral development. Academic Press) includes:
- What develops (or, what key dimensions develop, whether in individuals, relationships, cultural communities, or organizations)?
- What does development look like (descriptively) over time, in terms of both stability and change?
- What correlated factors predict these pathways of development?
- What causal factors enhance or impede development?
- How can we understand individual differences in development?
It would greatly strengthen this paper if the authors conducted such a developmental analysis on the developmental relational pathways of refugee children, and if they cited existing evidence and/or sketch needed studies to address the developmental issues they specify in their analysis.
Author Response
please see attachment - letter to editor and reviewers

Reviewer 2 Report
First of all, I would like to thank the authors for their work on the protection of refugee children, specifically on Ambivalence Towards the Protection of Refugee Children. Since the study is focusing on refugee protection, I miss general information on the international legal framework protecting refugees, the migration (health) cycle, and the public health relevance of this topic as my major concerns. Why is it important to focus on the protection of refugee children? How many children worldwide are affected? What are the (public) health consequences of the identified ambivalence in protecting refugee children? There are some important research findings on the mental health and wellbeing of refugee children that should be discussed in the context of the paper. Please also check on (IJERPH-)publications on family separation and mental health. A minor detail in line 182: UK is still part of Europe, but not of the European Union - why is UK listed separately? Overall, I find this an interesting paper that has potential, but in its current state is inappropriate for the International Journal of Environmental Research and Public Health.
Author Response

(The authors gave the same response as above.)

Round 2
Reviewer 2 Report
The authors have met all the issues I rose in reviewing the first version of the manuscript and I haven't found any additional issues in the new sections.